# Disruption of *Osteoprotegerin* has complex effects on medial destruction and adventitial fibrosis during mouse abdominal aortic aneurysm formation

Batmunkh Bumdelger◉, Mikage Otani◉, Kohei Karasaki, Chiemi Sakai, Mari Ishida, Hiroki Kokubo◉ *, Masao Yoshizumi *

Department of Cardiovascular Physiology and Medicine, Graduate School of Biomedical and Health Sciences, Hiroshima University, Hiroshima, Japan

◉ These authors contributed equally to this work.
* hkokubo@hiroshima-u.ac.jp (HK); yoshizum-tky@umin.ac.jp (MY)

## Abstract

Aortic aneurysm refers to dilatation of the aorta due to loss of elasticity and degenerative weakening of its wall. A preventive role for osteoprotegerin (Opg) in the development of abdominal aortic aneurysm has been reported in the $CaCl_2$-induced aneurysm model, whereas Opg was found to promote suprarenal aortic aneurysm in the AngII-induced *ApoE* knockout mouse aneurysm model. To determine whether there is a common underlying mechanism to explain the impact of Opg deficiency on the vascular structure of the two aneurysm models, we analyzed suprarenal aortic tissue of 6-month-old $ApoE^{-/-}Opg^{-/-}$ mice after AngII infusion for 28 days. Less aortic dissection and aortic lumen dilatation, more adventitial thickening, and higher expression of collagen I and Trail were observed in $ApoE^{-/-}Opg^{-/-}$ mice relative to $ApoE^{-/-}Opg^{+/+}$ mice. An accumulation of α-smooth muscle actin and vimentin double-positive myofibroblasts was noted in the thickened adventitia of $ApoE^{-/-}Opg^{-/-}$ mice. Our results suggest that fibrotic remodeling of the aorta induced by myofibroblast accumulation might be an important pathological event which tends to limit AngII-induced aortic dilatation in $ApoE^{-/-}Opg^{-/-}$ mice.

## Introduction

Aortic aneurysm (AA) refers to a dilatation of the aorta due to loss of elasticity and degenerative weakening of its wall. Continuous expansion of the aorta results in rupture and is associated with a high mortality rate [1]. Analyses of AA in experimental animal models, including the $CaCl_2$-induced mouse model and the AngII-induced *ApoE* knockout (KO) mouse model [2, 3], are important for understanding the pathogenesis of this disease and for developing effective drug treatments aimed at arresting aortic expansion [4–7].

Osteoprotegerin (Opg, also referred to as TNFRSF11B), a member of the tumor necrosis factor (TNF) receptor superfamily, functions as a decoy receptor to regulate various factors

**Data Availability Statement:** All relevant data are within the manuscript and its Supporting Information files.

**Funding:** Our study was supported by a Grant-in-Aid for Scientific Research from the Japanese Ministry of Education, Culture, Sports, Science, and Technology (18K07878) to MY.

**Competing interests:** The authors have declared that no competing interests exist.

across many biological processes [8]. For example, Opg has been shown to regulate bone metabolism through the Receptor activator of nuclear factor kappa-B ligand (Rankl) [9, 10] and apoptosis of cancer cells through TNF-related apoptosis-inducing ligand (Trail). Given that vascular diseases are often involved in bone pathologies [11, 12], and because Opg is expressed in vascular smooth muscle cells (VSMCs) [13] and serum levels of OPG are elevated in cardiovascular disease [14–17], there is great interest in the roles Opg may play in the vascular system.

We recently reported that Opg plays a preventive role in the development of abdominal AA (AAA) in the CaCl₂-induced aneurysm model [18]. In *Opg* KO mice, we found larger aneurysms with destruction of the aortic medial layer, which had increased expression of matrix metalloproteinase (Mmp)-9 and Trail. The expression of Opg in aortic tissue was also increased in response to aneurysm induction in wild-type mice. We concluded in this recent study that Opg prevents AAA formation through its antagonistic effect on Trail. However, another group reported that Opg can promote the development of aneurysms in the suprarenal aorta (SRA) in the AngII-induced *ApoE* KO mouse model [19]. The authors of that study attributed the lower incidence of aneurysms and rupture in $ApoE^{-/-}Opg^{-/-}$ mice to the downregulation of proteolytic enzyme expression. The two reports discussed above suggest opposing results for aneurysm formation in *Opg*-deficient mice. Interestingly, a recent study found that low dose Opg had a preventive effect in both AAA models, although the mechanism underlying the preventive effect is unclear [20].

In the present study, we found a lower incidence of aortic dissection and a lower tendency for aortic dilatation in *Opg*-deficient mice. There were no structural differences in medial elastic fibers between $ApoE^{-/-}Opg^{-/-}$ and $ApoE^{-/-}Opg^{+/+}$ mice. However, more adventitial thickening and increased expression of collagen I, α-smooth muscle actin (SMA), vimentin, and Trail were observed in $ApoE^{-/-}Opg^{-/-}$ mice relative to $ApoE^{-/-}Opg^{+/+}$ mice. This suggests that fibrotic remodeling of the aorta may have resulted from an accumulation of myofibroblasts in $ApoE^{-/-}Opg^{-/-}$ mice, which was also previously observed in the CaCl₂-induced AAA model [18]. Our results suggest that Opg deficiency may lead to fibrotic remodeling of the aorta, possibly enhanced by Trail signaling, and is an important pathological event that tends to limit AngII-induced SRA dilatation and dissection in *ApoE* KO mice.

## Results

### Opg deficiency tends to limit AngII-induced aortic dissection and dilatation

In order to confirm the phenotypic differences in the two aneurysm models resulting from *Opg* deficiency, we crossed *Opg* KO mice with *ApoE* KO mice to generate $ApoE^{-/-}Opg^{-/-}$ mice. The *Opg* KO mice were then treated with AngII to promote aneurysm development in the suprarenal aorta (SRA) (S1A Fig in S1 File). Although Moran et al. reported a significantly smaller median maximum diameter of the SRA in $ApoE^{-/-}Opg^{-/-}$ mice compared to that in $ApoE^{-/-}Opg^{+/+}$ mice, we did not observe a significant reduction in the maximum external diameter in $ApoE^{-/-}Opg^{-/-}$ mice relative to $ApoE^{-/-}Opg^{+/+}$ mice (Fig 1A and 1B, S2A Fig–S2C Fig in S1 File). Moreover, no significant differences were observed in survival rate and concentrations of serum cholesterol between $ApoE^{-/-}Opg^{-/-}$ and $ApoE^{-/-}Opg^{+/+}$ mice at 28 days after AngII infusion (S2F Fig and S2G Fig in S1 File). However, on visual inspection, the diameter of the SRA in $ApoE^{-/-}Opg^{-/-}$ mice appeared to be somewhat smaller, and multiple hematomas were observed in the SRA of $ApoE^{-/-}Opg^{+/+}$ mice. Thus, we adopted Daugherty's classification system for aortic aneurysms [21] and categorized the mice into three groups based on dilatation of the SRA and the presence of visible hematoma (No Aneurysm, Aneurysm, and

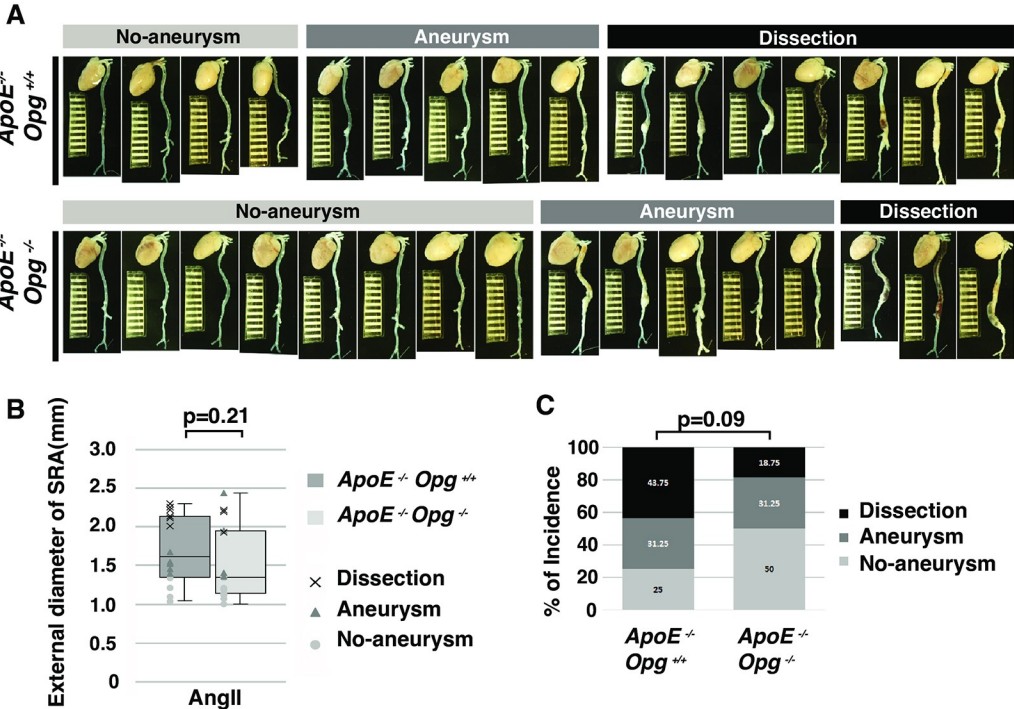

**Fig 1. Opg deficiency tends to suppress AngII-induced aortic aneurysms. (A)** Aortas of AngII-infused $ApoE^{-/-}Opg^{+/+}$ and $ApoE^{-/-}Opg^{-/-}$ mice were categorized into three groups based on diameter and the presence of visible hematoma (No Aneurysm, Aneurysm, and Dissection groups). Scale bars indicate 1 mm. **(B)** External diameter of the SRA in AngII-infused $ApoE^{-/-}Opg^{+/+}$ (gray, n = 16) and $ApoE^{-/-}Opg^{-/-}$ (white, n = 16) mice. All measurements are shown as box plots and each measurement is shown as a white circle (No Aneurysm), gray triangle (Aneurysm), or black cross (Dissection). **(C)** The incidence (%) of aortic aneurysms is shown for No Aneurysm (white), Aneurysm (gray), and Dissection (black) groups. Statistical significance: p<0.05.

Dissection groups). In $ApoE^{-/-}Opg^{+/+}$ mice, 75% of SRAs were dilated and the majority of them had dissected (Fig 1C). However, in $ApoE^{-/-}Opg^{-/-}$ mice, 50% of SRAs were dilated and only a few had dissected. This suggests that Opg deficiency may have a preventive effect on AngII-induced aortic dilatation and dissection (43.75% in $ApoE^{-/-}Opg^{+/+}$ mice vs. 18.75% in $ApoE^{-/-}Opg^{-/-}$ mice).

## Opg deficiency results in aneurysm with adventitial thickening

Since Opg deficiency may have a preventive effect on aortic dilatation and dissection, transverse cross-sections of the SRA were used to measure the size of the aortic lumen and evaluate the aortic wall structure, including the medial and adventitial layers. HE staining revealed narrower aortic lumens and smaller dissecting hematomas in $ApoE^{-/-}Opg^{-/-}$ mice compared to $ApoE^{-/-}Opg^{+/+}$ mice (Fig 2A(HE)). The area of the aortic internal lumen tended to be smaller in $ApoE^{-/-}Opg^{-/-}$ mice compared with $ApoE^{-/-}Opg^{+/+}$ mice, although the difference was not significant (Fig 2B).

There were no differences in the structure of medial elastic fibers in the Aneurysm group between $ApoE^{-/-}Opg^{-/-}$ and $ApoE^{-/-}Opg^{+/+}$ mice, as assessed by EVG and AZAN staining (Fig 2A (panels EVG and Aneurysm), S3A Fig in S1 File). In the Dissection group, a complete disappearance of medial elastic fibers in which atherosclerotic plaques invaded the disrupted side of the aortic circumference was observed in mice of both genotypes (Fig 2A (panels EVG and Aneurysm)). There was no significant difference in the width of the medial layer between mice

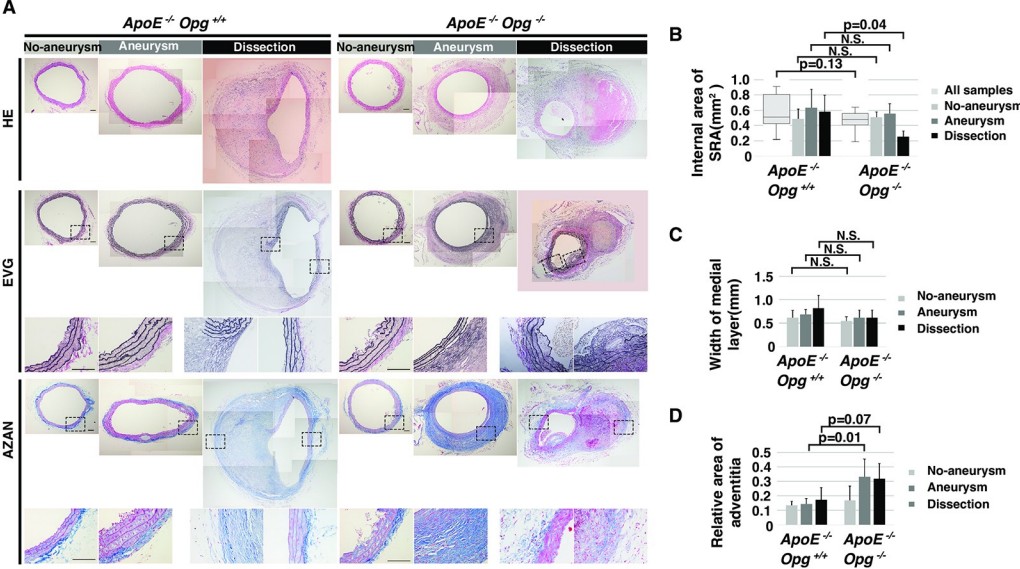

**Fig 2. Opg deficiency tends to suppress AngII-induced aortic dilatation and promotes adventitial thickening. (A)** Representative cross sections of the aorta stained with HE, EVG, and AZAN. Areas selected by boxes surrounded by a black dotted line are magnified below. Scale bars indicate 0.1 mm. **(B)** Internal area of the suprarenal aorta (SRA) of *ApoE<sup>-/-</sup>Opg<sup>+/+</sup>* (n = 16) and *ApoE<sup>-/-</sup>Opg<sup>-/-</sup>* (n = 16) mice. Measurements for all samples are presented as box plots. Measurements for the three groups are presented as bar graphs. n = 4 and 8 for the No-aneurysm group, n = 5 and 5 for the Aneurysm group, and n = 7 and 3 for the Dissection group in *ApoE<sup>-/-</sup>Opg<sup>+/+</sup>* and *ApoE<sup>-/-</sup>Opg<sup>-/-</sup>* mice, respectively. **(C)** Medial layer width of the SRA of AngII-infused mice in each group. N.S; not significant. **(D)** Relative adventitial area of the SRA of AngII-infused mice in each group. Statistical significance: p<0.05.

of both genotypes (Fig 2C). However, significant adventitial thickening and accumulation of collagen were noted in the Aneurysm and Dissection groups in *ApoE<sup>-/-</sup>Opg<sup>-/-</sup>* mice (Fig 2A (panel AZAN), Fig 2D, S3B Fig in S1 File). These observations suggest that adventitial thickening might be associated with the smaller aortic diameter and lower tendency of dissection in *ApoE<sup>-/-</sup>Opg<sup>-/-</sup>* mice.

### Fibrotic remodeling of the SRA and accumulation of myofibroblasts in *ApoE<sup>-/-</sup>Opg<sup>-/-</sup>* mice

Given the accumulation of collagen in *ApoE<sup>-/-</sup>Opg<sup>-/-</sup>* mice, we examined the type of collagen that accumulated in the adventitia. Collagen I, but not collagen III, accumulated in *ApoE<sup>-/-</sup>Opg<sup>-/-</sup>* mice following AngII infusion (Fig 3A, S4A Fig and S4B Fig in S1 File). The area of collagen I expression was significantly larger in *ApoE<sup>-/-</sup>Opg<sup>-/-</sup>* mice compared to that in *ApoE<sup>-/-</sup>Opg<sup>+/+</sup>* mice (Fig 3B). As expected, no differences were found in collagen I expression between *ApoE<sup>-/-</sup>Opg<sup>-/-</sup>* and *ApoE<sup>-/-</sup>Opg<sup>+/+</sup>* mice in the $H_2O$-infused controls (S4C Fig in S1 File). Since TGF-β1 is the central mediator of fibrogenesis [22], we examined its mRNA expression in SRA tissue and found it to be significantly higher in *ApoE<sup>-/-</sup>Opg<sup>-/-</sup>* mice compared to *ApoE<sup>-/-</sup>Opg<sup>+/+</sup>* mice after AngII infusion (Fig 3C). This result further supports the finding that Opg deficiency promotes fibrotic remodeling of the aorta.

To determine the types of cells which accumulated in the adventitia, immunohistochemistry was performed using anti-SMA and anti-vimentin antibodies to mark smooth muscle cells and fibroblasts, respectively. In *ApoE<sup>-/-</sup>Opg<sup>-/-</sup>* mice, both SMA and vimentin were distributed in the adventitia, especially in the Aneurysm and Dissection groups (Fig 4A and 4B). Since both SMA and vimentin are known to be expressed in myofibroblasts, the accumulated cells

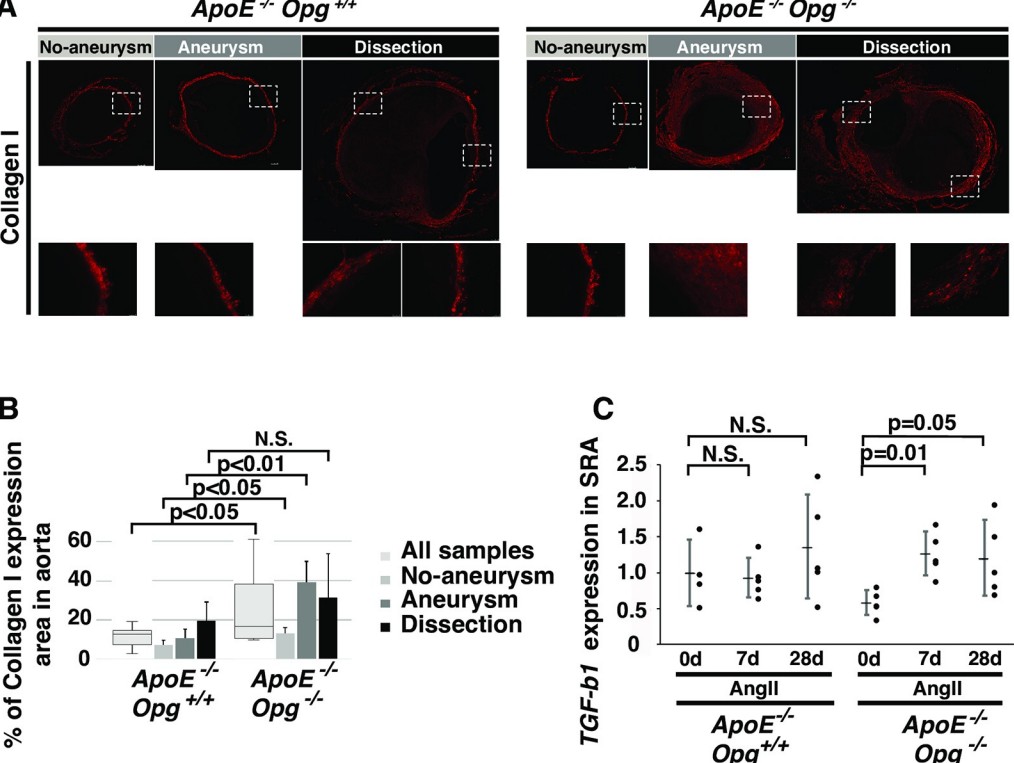

**Fig 3. Collagen accumulation in aortas of *ApoE*⁻/⁻*Opg*⁻/⁻ mice. (A)** Representative immunofluorescence images of aortas of AngII-infused mice stained with an anti-collagen I antibody. Boxed areas (with white dotted lines) are magnified below. Scale bars indicate 100 μm. **(B)** Percent area of collagen I expression in aortas of *ApoE*⁻/⁻*Opg*⁺/⁺ (n = 16) and *ApoE*⁻/⁻*Opg*⁻/⁻ (n = 16) mice. Measurements for all samples are presented as box plots. Measurements for the three groups are presented as bar graphs (n = 4 and 8 for the No Aneurysm group, n = 5 and 5 for the Aneurysm group, and n = 7 and 3 for the Dissection group in *ApoE*⁻/⁻*Opg*⁺/⁺ and *ApoE*⁻/⁻*Opg*⁻/⁻ mice, respectively). **(C)** *Tgf-β1* mRNA expression in the SRA at days 0, 7, and 28 after initiation of AngII infusion in *ApoE*⁻/⁻*Opg*⁺/⁺ (n = 4, 5, 5) and *ApoE*⁻/⁻*Opg*⁻/⁻ (n = 5, 5, 5) mice. Statistical significance: p<0.05.

that over-express collagen I in the adventitia of *ApoE*⁻/⁻*Opg*⁻/⁻ mice are likely myofibroblasts. Interestingly, these myofibroblasts were found not only in the adventitia, but also in the outer layers of the media in *ApoE*⁻/⁻*Opg*⁻/⁻ mice. Medial cells expressing SMA but not vimentin in *ApoE*⁻/⁻*Opg*⁺/⁺ mice were unlikely to be myofibroblasts (Fig 4A). Next, we measured the expression of matrix metalloproteinases (MMPs) in SRA tissue, but found no difference between *ApoE*⁻/⁻*Opg*⁻/⁻ and *ApoE*⁻/⁻*Opg*⁺/⁺ mice after AngII infusion (S5A Fig-S5C Fig in S1 File). These findings collectively suggest that accumulation of myofibroblasts could be one of the causes of adventitial thickening in *ApoE*⁻/⁻*Opg*⁻/⁻ mice after AngII infusion.

## Up-regulation of Trail in SRAs of AngII-infused *ApoE*⁻/⁻*Opg*⁻/⁻ mice

To understand how Opg deficiency contributes to adventitial thickening, we examined the expression of Trail in aneurysm tissue of AngII-infused *ApoE*-KO mice, given the known role of Opg as a decoy receptor for Trail [8]. Trail expression was up-regulated in the adventitia and outer layers of the media in *ApoE*⁻/⁻*Opg*⁻/⁻ mice after AngII infusion (Fig 5A and 5C), but not in *ApoE*⁻/⁻*Opg*⁺/⁺ mice or after H₂O infusion (S5D and S5F Fig in S1 File). Notably, Trail was mainly expressed in cells that were double positive for SMA and vimentin. These cells

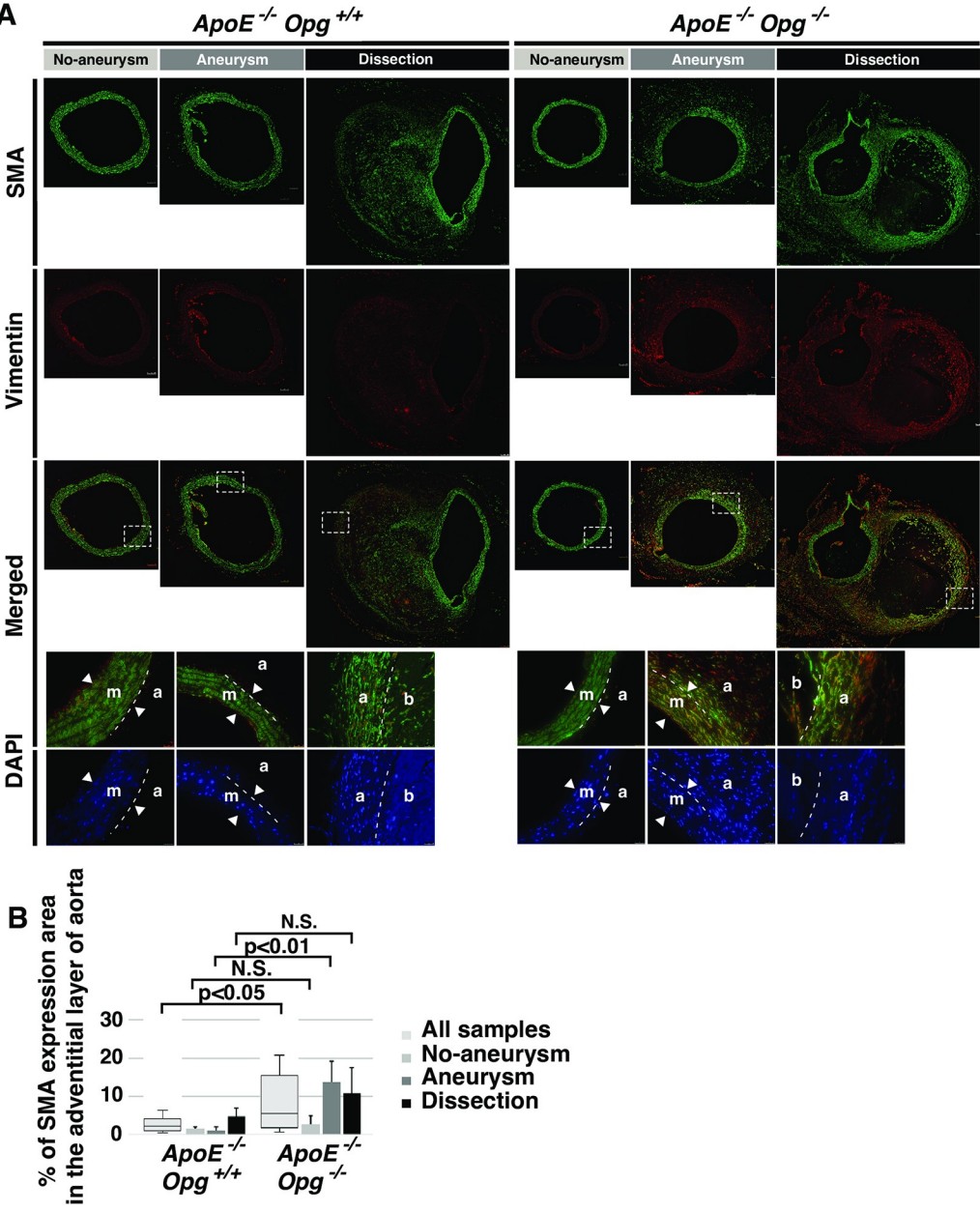

**Fig 4. Accumulation of myofibroblasts in adventitias of AngII-infused *ApoE⁻/⁻Opg⁻/⁻* mice. (A)** Representative double-immunofluorescence images of aortas after AngII infusion with anti- SMA (green) and anti-vimentin (red) antibodies. Scale bars indicate 100 μm. Areas selected by the white box (dotted line) are magnified below. Nuclei were stained with DAPI (blue). Scale bars indicate 25 μm. Medial layer (m), Adventitial layer (a), hematoma (b). The two arrowheads show borders of the medial layer. The white dotted line in the magnified panels indicates the border of the medial and adventitial layers. **(B)** Percent area of SMA expression in adventitias of *ApoE⁻/⁻Opg⁺/⁺* (n = 16) and *ApoE⁻/⁻Opg⁻/⁻* (n = 16) mice. Measurements for all samples are presented as box plots. Measurements for the three groups are presented as bar graphs. n = 4 and 8 for the No Aneurysm group, n = 5 and 5 for the Aneurysm group, and n = 7 and 3 for the Dissection group in *ApoE⁻/⁻Opg⁺/⁺* and *ApoE⁻/⁻Opg⁻/⁻* mice, respectively. Statistical significance: $p < 0.05$.

were also positive for Ki67, a cell proliferation marker (Fig 5B), implying that Trail induces the proliferation of these cells autonomously [23].

Trail has also been reported to promote chemotactic migration of not only SMCs, but also monocytes towards the site of inflammation [24]. Round shaped, F4/80 macrophages were

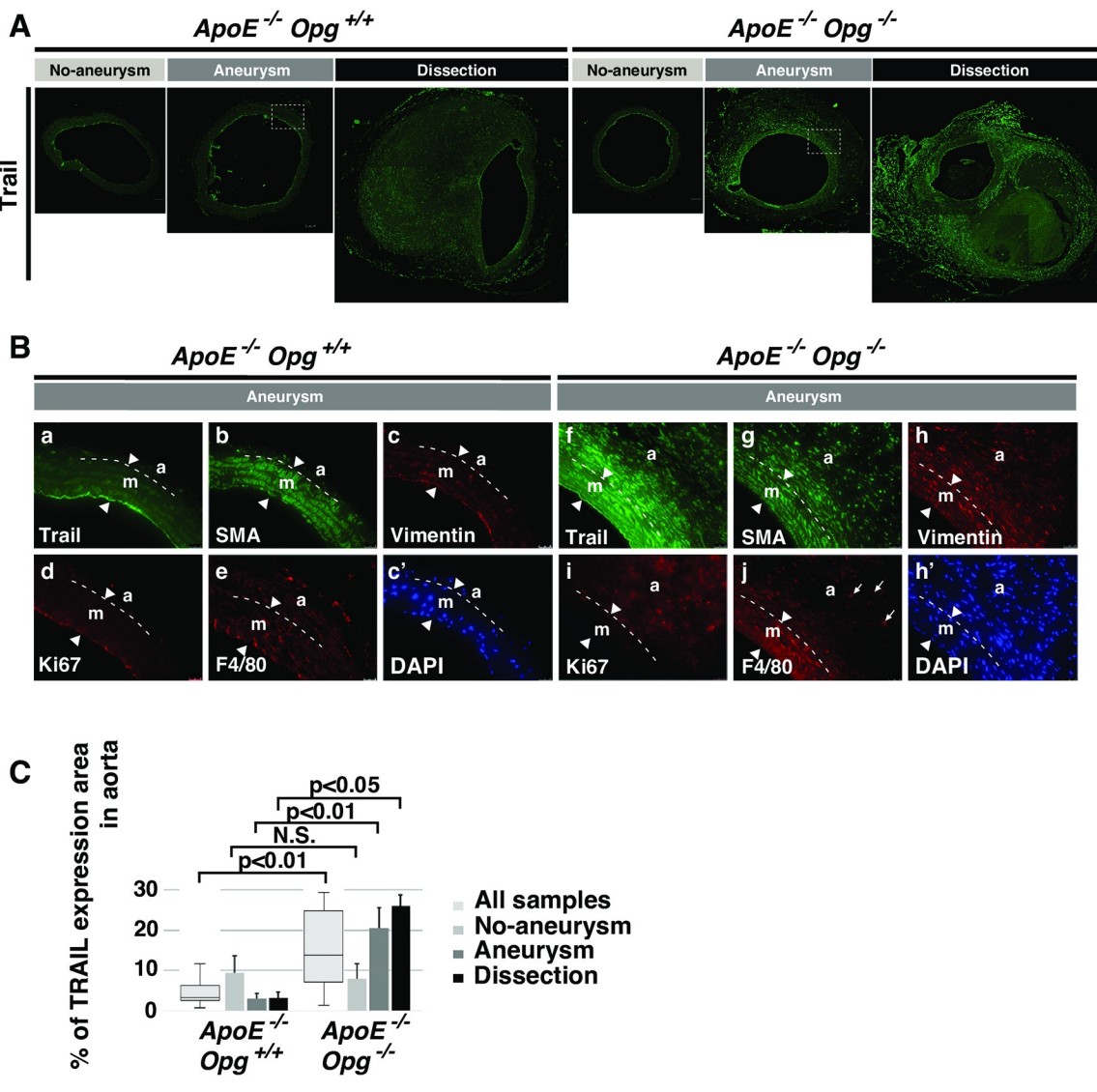

**Fig 5. Trail upregulation in aneurysm tissue of AngII-infused *ApoE⁻/⁻Opg⁻/⁻* mice. (A)** Representative immunofluorescence images of aortic tissue sections from AngII-infused mice stained with anti-Trail (green) antibody. Scale bars indicate 100 μm. Areas selected by the white box (dotted line) in the Aneurysm group are magnified below. **(B)** Magnified immunofluorescence images of aortic tissue sections from the AngII-infused Aneurysm group stained with anti-Trail (green; a, f), anti-SMA (green; b, g), anti-vimentin (red; c, h), anti-Ki67 (red; d, i), and anti-F4/80 (red; e, j) antibodies in *ApoE⁻/⁻Opg⁺/⁺* and *ApoE⁻/⁻Opg⁻/⁻* mice. Nuclei were stained with DAPI (blue; c-DAPI, h-DAPI). Scale bars indicate 25 μm. Medial layer (m), adventitial layer (a). The two arrowheads show borders of the medial layer. The white dotted line in the magnified panels indicates the border of the medial and adventitial layers. Small arrows in panel j indicate large round-shaped macrophages infiltrating the adventitia. Statistical significance: p<0.05. **(C)** Percent area of Trail expression in medial and adventitial layers of *ApoE⁻/⁻Opg⁺/⁺* (n = 16) and *ApoE⁻/⁻Opg⁻/⁻* (n = 16) mice. Measurements for all samples are presented as box plots. Measurements for the three groups are presented as bar graphs. n = 4 and 8 for the No Aneurysm group, n = 5 and 5 for the Aneurysm group, and n = 7 and 3 for the Dissection group in *ApoE⁻/⁻Opg⁺/⁺* and *ApoE⁻/⁻Opg⁻/⁻* mice, respectively. Statistical significance: p<0.05.

detected in the adventitia of *ApoE⁻/⁻Opg⁻/⁻* mice (Fig 5B; S5G Fig in S1 File), suggesting that increased Trail expression may lead to more inflammation, myofibroblast accumulation, and fibrotic remodeling of the aortic wall in Opg-deficient mice.

## Discussion

Consistent with results reported by Moran et al. [19], we found that Opg deficiency tended to limit AngII-induced aortic dissection and dilatation. Gross phenotypic differences are evident

between the CaCl$_2$-induced aneurysm model and AngII-induced aneurysm model [18, 19]. However, we observed adventitial thickening accompanied by myofibroblast accumulation and increased expression of collagen I and Trail in aneurysm tissue of *ApoE$^{-/-}$Opg$^{-/-}$* mice. These pathological and cellular changes of the aortic wall were also reported in the CaCl$_2$-induced aneurysm model [18], suggesting that adventitial thickening with myofibroblast accumulation could be a common feature of Opg deficiency.

Myofibroblasts in aortic tissue promote the excessive production of extracellular matrix, including collagen, and subsequent fibrotic remodeling of the aorta [25]. Disorders of collagen fiber assembly are thought to underlie aortic dilatation and dissection. For example, weakness in connective tissue caused by mutations in various types of collagen and enzymes involved in collagen maturation in Ehlers-Danlos syndrome results in aortic aneurysms [26]. Moreover, Dobrin et al. reported that degradation of collagen, rather than elastin, is an important cause of aneurysm development, suggesting that collagen fibers protect the aorta from internal pressure-induced expansion [27]. When applied to the present study, these findings suggest that excessive collagen accumulation in the adventitia may have strengthened the aortic tissue. This in turn could have resulted in smaller aneurysms and a lower rate of dissection events in *ApoE$^{-/-}$Opg$^{-/-}$* mice in the AngII-induced aneurysm model. In light of studies reporting that Trail induces collagen I transcription in fibroblast cells [28], Trail could potentially play a role in collagen accumulation in myofibroblasts. Fibrotic remodeling of the aorta by myofibroblasts might protect against AngII-induced aneurysm and dissection in *ApoE$^{-/-}$Opg$^{-/-}$* mice.

In our previous study, which used the CaCl$_2$-induced aneurysm model, Opg deficiency exacerbated AAAs, despite adventitial thickening, increased Trail expression, and the appearance of myofibroblasts [18]. We speculate that the contrasting phenotypes of the AngII-induced and CaCl$_2$-induced aneurysm models could be due to different processes involved in aneurysm formation. In the AngII-induced *ApoE* KO mouse aneurysm model, local medial destruction is observed only in the plaque area. It is currently thought that aneurysms and dissection start at small cleavages which develop after infusion of AngII, leading to local destruction of the medial layer [29]. Concurrently, collagen accumulation in the adventitia induced by AngII infusion in *ApoE$^{-/-}$/Opg$^{-/-}$* mice limits the development of aneurysms and dissection. On the other hand, in the CaCl$_2$-induced model, complete destruction of the medial layer is observed in a broad area. This extensive destruction of the medial layer may promote enlargement of the aortic lumen. Increased Trail expression, due to the absence of Opg, up-regulates the expression of proteolytic enzymes, including Mmp9 and Mmp2, resulting in an acceleration of elastic medial tissue destruction. Although adventitial thickening simultaneously progresses after CaCl$_2$ treatment in *Opg* KO mice, it may be insufficient to compensate for the severe and extensive medial tissue destruction and limit aortic enlargement.

In conclusion, adventitial thickening with collagen accumulation induced by AngII infusion may increase the strength of aortic tissue and potentially limit the development of aortic aneurysms and dissection in *ApoE$^{-/-}$Opg$^{-/-}$* mice. At the same time, aortic tissue may lose flexibility in elastic vessels, thereby increasing cardiac load. AngII infusion in the absence of Opg may promote excessive reactions in aortic tissue, suggesting that Opg plays a potential role in aortic tissue homeostasis and maintenance of proper blood pressure. In the therapeutic context, targeting aortic tissue with agents that induce adventitial thickening with collagen accumulation may help suppress dissection events. Such treatments may include focal drug delivery systems in which an anti-Opg antibody is delivered to plaques, or direct application of drugs to stent grafts which would allow for focal induction of adventitial thickening.

## Materials and methods

### Mice

*Opg* KO ($ApoE^{+/+}/Opg^{-/-}$) mice of the C57BL/6J strain background (CLEA Japan, Inc.) were crossed with *ApoE* KO ($ApoE^{-/-}/Opg^{+/+}$) mice to obtain $ApoE^{+/-}Opg^{+/-}$ mice. Progeny were inter-crossed to establish an $ApoE^{-/-}Opg^{+/-}$ line, which was then inter-crossed to generate $ApoE^{-/-}Opg^{-/-}$ mice and control $ApoE^{-/-}Opg^{+/+}$ mice. Genotypes were determined by PCR (94˚C for 30 sec; 55˚C for 30 sec and 72˚C for 30 sec for 35 cycles) using the following primers: forward primer 5'-CTG ACC ACT CTT ATA CGG AC AG-3' and reverse primer 5'-CTA AGT TAG CTG CTG TCT GGC-3' for the *Opg*-wild type allele; forward primer 5'-CTG ACC GCT TCC TCG TGC TTTAC-3' and the above-mentioned reverse primer for the *Opg*-mutant allele; forward primer 5'-ACT CTA CAC AGG ATG CCT AGC-3' and reverse primer 5'-CTC ACG TCA GTT CTT GTG TGAC-3' for the *ApoE*-wild type allele; and the above-mentioned forward primer and reverse primer 5'-GCC GCC CGA CTG CAT CT-3' for the *ApoE*-mutant allele.

### Aortic aneurysm model

Six-month-old male $ApoE^{-/-}Opg^{+/+}$ (n = 16) and $ApoE^{-/-}Opg^{-/-}$ (n = 16) mice were infused with AngII as described previously [30]. Under anesthesia, a micro-osmotic pump (ALZET Model 1004, Durect Corporation) was implanted into the subcutaneous space left of the dorsal midline for infusion of AngII (Sigma-Aldrich) or $H_2O$ at a rate of 1.0 μg/kg/min for 28 days. Mice were sacrificed by anesthetia overdose. After cutting the sternum and exposing the thoracic cavity, blood from the right ventricle was collected for storage during perfusion with PBS. Mice were then perfused with 4% paraformaldehyde. Whole aortas were excised for morphometric analysis and measurement of SRA diameters. Images were taken of whole aortas to measure the external diameter of SRAs. SRAs were then separated from aortas and embedded in paraffin for subsequent histological analysis.

For RNA isolation, aortas were harvested after perfusion with PBS and stored in RNA-later solution (Ambion). This experiment was approved by the Committee of Animal Experimentation at Hiroshima University (A08-32) and carried out in accordance with the approved protocol. All surgeries were performed under a combination of three anesthetic solutions (medetomidine, butorphanol, and midazolam), and efforts were made to minimize suffering during and after surgery.

### Morphological examination and measurement of aortic diameters

The maximum external diameter of the SRA as shown in S1A Fig in S1 File was measured (Fig 1B). The external diameter of the SRA in the $H_2O$-infused group (S2C Fig in S1 File) was used as a control to categorize AngII-infused aortas into three groups based on maximum external diameter and the presence of visible hematoma (No-aneurysm group: diameter less than 1.5 times the $H_2O$-infused control; Aneurysm group: diameter more than 1.5 times the $H_2O$-infused control without visible hematoma; Dissection group: diameter more than 1.5 times the $H_2O$-infused control with visible hematoma). Paraffin-embedded aortic tissues were used to generate 6 μm-thick sections, which were cut at the level of the SRA. The tissue sections were then subjected to Hematoxylin/Eosin (HE), Elastica van Gieson (EVG), and Analine Blue (AZAN) staining. The sections were used to measure the internal area of the SRA, the width of the medial layer, and the relative area of the adventitia. Borders of the aortic lumen were marked with a dotted circle (S1B Fig in S1 File) and used to measure the internal area of the SRA (Fig 2B). Red lines show 5–8 measurements of the medial layer width on the side of the

aortic circumference which was not disrupted by atherosclerotic plaques (S1B Fig in S1 File). Average values were used (Fig 2C). The total area (TA) of the aorta included all aortic layers. The arterial area (AA), which included the lumen, endothelium, and media, and the bleeding area (BA), which included hematoma between medial and adventitial layers, were marked by dotted circles (S1C Fig and S1D Fig in S1 File). After measuring each area, the relative area of the adventitia was calculated using the following equation: TA-AA-BA/TA. The % area of collagen I expression was calculated by dividing the signal area by the entire aortic area (excluding the hematoma area) (Fig 3B). The % area of SMA expression was calculated by dividing the signal area by the area of the adventitial layer (Fig 4B). The % area of Trail expression was calculated by dividing the signal area by the area of medial and adventitial layers (Fig 5C). Area selection was performed using Photoshop software and was confirmed by three independent researchers. Area measurements were performed using Image-J software (NIH, USA).

## Immunohistochemistry (IHC)

Sections were pre-incubated in antigen retrieval solution (pH 5.2) at 90˚C for 45 minutes, based on the manufacturer's instructions (Dako), and then blocked with 1% bovine serum albumin in 0.1% Tween-phosphate buffered solution (PBS) for 1 hr. After pre-treatment and blocking, sections were incubated with a primary antibody overnight at 4˚C and subsequently with appropriate secondary antibodies for 2 hr at room temperature, followed by counterstaining with 4'-6-diamidino-2-phenylindole (DAPI). After primary antibodies, listed in Table 1, sections were treated with anti-mouse, -rat, or -rabbit antibodies conjugated wiith Alexa Fluor 488 or 555 dye (donkey, 1:500; Life Technologies). Signals were detected using a DMI4000 fluorescence microscope (Leica Microsystems).

## Quantitative Real-time PCR

Total RNA was isolated using TRIzol reagent (Invitrogen). Reverse transcription was performed using the ReverTra Ace qPCR RT Kit (TOYOBO). Real-time PCR was conducted using SYBR Premix Ex Taq II (Takara Bio Inc. and Kapa Biosystems Inc.). Intensities of PCR products using primers, listed in Table 2, were measured and analyzed using Opticon (MJ Research). Amplification conditions were as follows: 5 s at 95˚C, 20 s at 60˚C, and 15 s at 72˚C for 49 cycles. *G3pdh* was used as the internal control.

## Serum cholesterol measurement

Concentrations of serum cholesterol were measured in $ApoE^{-/-}Opg^{+/+}$ and $ApoE^{-/-}Opg^{-/-}$ mice on days 0, 7, and 28 after initiation of AngII infusion. Measurements were performed by JaICA, Nikken Seil Co., Ltd (Shizuoka, Japan).

**Table 1. List of antibodies.**

| Specificity | Vendor | Cat# | Lot# | dilution |
|---|---|---|---|---|
| Collagen I | Abcam | ab21286 | GR46228-1 | 1:250 |
| Collagen III | Abcam | ab7778 | GR52659-1 | 1:250 |
| TRAIL | Abcam | ab2435 | GR14018-5 | 1:50 |
| Actin, α-Smooth Muscle (Monoclonal clone 1A4) | Sigma-Aldrich | A2547 | 032M4822 | 1:200 |
| Vimentin | BioVision | 3634 | | 1:50 |
| Ki67 | Abcam | ab15580 | GR101835-1 | 1:100 |
| F4/80 (BM8) | Santa Cruz | sc-52664 | B1810 | 1:50 |

**Table 2. List of PCR primers.**

| Genes | Sequences | Product size (bp) |
|---|---|---|
| *Mmp-9*: | forward 5' -GCCCTGGAACTCACACGACA-3' | 85 |
| | reverse 5'-TTGGAAACTCACACGCCAGAAG-3' | |
| *Mmp-2*: | forward 5'-CTCCTACAACAGCTGTACCAC-3' | 182 |
| | reverse 5'-CATACTTGTTGCCCAGGAAAG-3' | |
| *Tgf-β1*: | forward 5'- ATCGACATGGAGCTGGTGAAA-3' | 76 |
| | reverse 5'- TGGCGAGCCTTAGTTTGGA-3' | |
| *G3pdh*: | forward 5'- ACCACAGTCCATGCCATCAC-3' | 452 |
| | reverse 5'- TCCACCACCCTGTTGCTGTA-3' | |

## Statistical analysis

Non-parametric analyses, including the Mann-Whitney U-test or Kruskal-Wallis test with Scheffe's and Steel-Dwass post hoc analyses, and ordinal logistic regression analyses were conducted using Ekuseru-Toukei 2012 software (Social Survey Research Information Co., Ltd.). Data are expressed as mean ± standard deviation (SD). $P<0.05$ was considered statistically significant.

## Supporting information

**S1 File.**
(PDF)

## Acknowledgments

We thank Mr. Masayoshi Takatani and other members in the Radiation Research Center for Frontier Science, Institute for Radiation Biology and Medicine at Hiroshima University for their technical assistances and Dr. Kenichi Satoh in the Center for Data Science Education and Research at Shiga University for valuable advice in the statistical analysis. We also appreciate for all member the Natural Science Center for Basic Research and Development (N-BARD) at Hiroshima University.

## Author Contributions

**Conceptualization:** Batmunkh Bumdelger, Chiemi Sakai, Mari Ishida, Hiroki Kokubo, Masao Yoshizumi.

**Funding acquisition:** Masao Yoshizumi.

**Investigation:** Batmunkh Bumdelger, Mikage Otani, Kohei Karasaki.

**Methodology:** Batmunkh Bumdelger, Hiroki Kokubo.

**Project administration:** Batmunkh Bumdelger, Hiroki Kokubo.

**Supervision:** Hiroki Kokubo, Masao Yoshizumi.

**Validation:** Chiemi Sakai, Mari Ishida, Masao Yoshizumi.

**Writing – original draft:** Batmunkh Bumdelger, Mikage Otani, Kohei Karasaki.

**Writing – review & editing:** Mikage Otani, Kohei Karasaki, Hiroki Kokubo, Masao Yoshizumi.

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
