## [Decision Letter · Decision Letter 0]

26 Mar 2020

PONE-D-20-04332

Disruption of Osteoprotegerin has complex effects on medial destruction and adventitial fibrosis during mouse abdominal aortic aneurysm formation

PLOS ONE

Dear Dr. Kokubo,

Thank you for submitting your manuscript to PLOS ONE. Based on careful evaluation by three expert reviewers, it is considered to have merit but does not fully meet PLOS ONE’s publication criteria as it currently stands. Therefore, we invite you to submit a revised version of the manuscript that addresses the points raised during the review process.

We would appreciate receiving your revised manuscript by May 10 2020 11:59PM. To enhance the reproducibility of your results, we recommend that if applicable you deposit your laboratory protocols in protocols.io, where a protocol can be assigned its own identifier (DOI) such that it can be cited independently in the future. For instructions see: http://journals.plos.org/plosone/s/submission-guidelines#loc-laboratory-protocols

A rebuttal letter that responds to each point raised by the academic editor and reviewer(s).  The revised text should be copied to this letter and its location (page and line numbers) indicated. This letter should be uploaded as separate file and labeled 'Response to Reviewers'.A marked-up copy of your manuscript that highlights changes made to the original version. This file should be uploaded as separate file and labeled 'Revised Manuscript with Track Changes'.An unmarked version of your revised paper without tracked changes. This file should be uploaded as separate file and labeled 'Manuscript'.

We look forward to receiving your revised manuscript.

Kind regards,

Helena Kuivaniemi, MD, PhD

Academic Editor

PLOS ONE

Journal Requirements:

"Our research was is supported by Grants‐in‐aid for Scientific Research (18K07878).".

i) Please provide an amended statement that declares *all* the funding or sources of support (whether external or internal to your organization) received during this study, as detailed online in our guide for authors at http://journals.plos.org/plosone/s/submit-now.  Please also include the statement “There was no additional external funding received for this study.” in your updated Funding Statement.

ii) Please include your amended Funding Statement within your cover letter. We will change the online submission form on your behalf.

4. Please include the method of sacrifice in the Methods section of your manuscript.

6. ** Please ensure that you include a title page within your main document ** (We note you have uploaded your title page in your cover letter). You should list all authors and all affiliations as per our author instructions and clearly indicate the corresponding author.

Reviewers' comments:

Reviewer's Responses to Questions

**Comments to the Author**

1. Is the manuscript technically sound, and do the data support the conclusions?

Reviewer #1: Partly

Reviewer #2: Partly

Reviewer #3: No

2. Has the statistical analysis been performed appropriately and rigorously? 

Reviewer #1: Yes

Reviewer #2: Yes

Reviewer #3: N/A

3. Have the authors made all data underlying the findings in their manuscript fully available?

Reviewer #1: Yes

Reviewer #2: Yes

Reviewer #3: Yes

4. Is the manuscript presented in an intelligible fashion and written in standard English?

Reviewer #1: Yes

Reviewer #2: No

Reviewer #3: Yes

5. Review Comments to the Author

Reviewer #1: Main summary

In the paper by Bumdelger et al, the authors evaluate the role of aneurysm/dissection in the Apoe-/- osteoprotegerin (Opg)-knockout mouse model. Interestingly, the authors introduce that using the calcium-chloride model, that deficiency of Opg had larger aneurysms and increased expression of matrix metalloproteinase-9 (MMP9) and TNF-related apoptosis-inducing ligand (TRAIL). Within this paper, using 7-month-old mice, AngII-infusion into Apoe-/- Opg-/- male mice resulted in lower external abdominal aortic diameter and lower incidence of aneurysms and dissections. The group goes on to do histological analysis and finds that collagen I expression is higher in the Apoe-/- Opg-/- mice under AngII-infusion, however this did not occur under vehicle conditions (water-infused, Supplemental Fig. 3). The group looks at alpha-smooth muscle actin, vimentin, TRAIL, F4/80, and collagen I and find that within mice that experience an aneurysm, Apoe-/- Opg-/- have increased expression of actin, vimentin, TRAIL, and collagen I that results in fibrotic remodeling that may help to lower the possibility of developing a dissection or aortic rupture. While the studies are interesting, there are some details that need to be addressed for these experiments.

Major concerns

1. If I understand correctly, both with calcium-chloride and the AngII-infusion model, TRAIL levels go up significantly. The authors state that TRAIL goes up, because the decoy receptor (Opg) is no longer expressed. If this is the case, then immunostaining for TRAIL should also be increased in the water-infused mice. If TRAIL expression is only tied to Opg, then the reader would benefit from knowing this in the water-infused mice. This could be placed in the supplement and help with the overall understanding of TRAIL within these models.

2. The authors do a good job of separating out non-aneurysm from aneurysm and dissection. In Figures 1-3, the authors examine all disease types, however in Figure 4, only aneurysm tissue is examined. Is it possible to look at non-aneurysm and dissection tissues for alpha-actin, vimentin, F4/80 and collagen I? It might help to keep it all consistent from figure to figure. There was also quantification for Figure 3, but none for Figure 4. Is it possible to do this for only 1 of the 4 proteins observed? Maybe vimentin or smooth muscle actin since quantification has already been done for collagen I.

3. Have the authors evaluated collagen IV levels in the aortic basal lamina? Is it increased in the Opg-knockout similar to collagen I?

4. It might be interesting to look at MMP2 and MMP9 levels in these Opg-knockout mice. My guess is that this might be lower in the Opg-knockout mouse, but if there is excessive remodeling off the tissue, then maybe not? This could be explored at the mRNA level if needed. Methods indicate that mRNA was collected, however I could not find any data concerning this.

Minor concerns

1. Was any ultrasound data collected? If it was collected, then placing this within the manuscript would benefit the reader.

2. Methods state that the mice were aged out to 6 months, however abstract indicates 7 months. Which is correct?

3. Suprarenal aorta (SRA) appears in the Introduction first and should be defined there.

4. “Revealed” is misspelled in the Introduction.

5. Statistical analysis should be done on the incidence data in Figure 1.

6. Did the authors measure any plasma/serum measures in the mice. Since these mice would be hypercholesterolemic, did the authors look at serum cholesterol levels?

7. Did the authors look at TGF-beta or FGF in the aorta? Since the authors conclude that fibrosis remodeling has occurred in the Opg-knockout, then measurement of these factors could help to enhance the overall conclusion.

8. There doesn’t seem to be any rupture information within these studies. Can the authors report on how many mice were lost due to aortic rupture in the wildtype and knockout mice?

Reviewer #2: The paper entitled "Disruption of Osteoprotegerin has complex effects on medial destruction and adventitial fibrosis during mouse abdominal aortic aneurysm formation" by Bumdelger et al examined the effect of osteoprotegerin (Opg) in abdominal aneurysm. While there is already conflicting data published on the role of Opg on aneurysm: one by the same authors concluding preventive role of Opg in CaCl2-induced AAA and another by Moran et al 2014 concluding disease promoting role of Opg in AngII-induced AAA; this paper attempts to confirm the role of Opg in AngII-induced mouse model of AAA. However, at current state of the paper, many conclusions are not fully supported by the data provided. I have following concerns:

1) Figure 1: The title of Figure 1 and conclusion "Opg deficiency limits AngII-induced aortic dissection and dilatation" is overstated. External diameter is not significantly reduced with Opg deficiency. How do the authors define dissection? The authors need to classify aneurysm based on Daugherty's classification (PMID: 11606327). Were they abdominal or thoracic dissection? The data is not statistically analyzed to say preventive effect of Opg deficiency on aortic dissection. What about the mortality rates and mortality data in these experimental mice?

2) Figure 2: I could not understand the mechanism behind adventitial thickening and aortic dissection. Do the authors have any speculation on how does the adventitial thickening leads to smaller aortic diameter and dissection in ApoE-/-Opg-/- mice?

3) Figure 3: What about collagen III? Picrosirius red staining needs to be performed for differential staining of collagen I vs collagen III.

4) Figure 4. The authors show that cells accumulated in adventitial region of Apoe-/-Opg-/- mice are likely myofibroblasts since they co-express SMA and vimentin. However, the Figure 4I shows more infiltration of F4/80+ macrophages in Apoe-/-Opg-/- compared to Apoe-/- alone in 4F. And it is surprising that there are fewer macrophages and less adventitial thickening in Apoe-/- mice in 4F and 4E. How does increased Trail expression, increased inflammation, myofibroblast accumulation and fibrotic remodeling can be protective to aortic dilation and dissection? The data presented and conclusion do not match. Instead, these data imply that there would be more disease in Apoe-/-Opg-/- mice because of more inflammation.

5) Why 6-months (24 weeks) old mice are used for the aneurysm studies? Aneurysm studies are best performed in 8-12 weeks old mice.

6) Poor quality of writing: There are several grammatical, typo and spelling errors in the sentences. The paper needs significant English editing.

Reviewer #3: The paper entitled " Disruption of Osteoprotegerin has complex effects on medial destruction and adventitial fibrosis during mouse abdominal aortic aneurysm formation" by Kokubo et al concluded that fibrotic remodeling of the aorta induced by myofibroblast accumulation might be an important pathological event which limits AngII-induced aortic dilatation in ApoE -/- Opg -/- mice.

The conclusion of the manuscript is not substantially based upon the experimental data and is highly speculative.

Specifically:

1. The statement ‘Opg deficiency limits AngII-induced aortic dissection and dilatation’ is an overstatement of the data. The AAA definition is missing and also lacks statistical analysis.

2. No evidence to show the specificity of antibodies to detect vSMCs and fibroblasts. Single IHC may not be sufficient to draw the conclusions.

3. The interpretation that ‘Opg deficiency was found to limit AngII-induced aortic dissection and dilatation’ is not convincing.

4. There are no reports/data to show that fibrotic remodeling of the aorta might protects against AAA.

5. Why the authors used 6 month old mice?

6. Since aortic dissection in these mice occur at early stage of the disease (4-10 days), determination of visible blood at day 28 may not be a good indicator for aortic dissection.

7. Please check for grammatical errors.

6. PLOS authors have the option to publish the peer review history of their article (what does this mean?). If published, this will include your full peer review and any attached files.

Reviewer #1: No

Reviewer #2: No

Reviewer #3: Yes: Chetan P Hans

---

## [Author Response · Author response to Decision Letter 0]

10 May 2020

Re: PONE-D-20-0433

We thank the Reviewers for the helpful and constructive comments on our manuscript, entitled “Disruption of Osteoprotegerin has complex effects on medial destruction and adventitial fibrosis during mouse abdominal aortic aneurysm formation.” We have considered the Reviewers’ comments carefully and have revised the manuscript to address their concerns and questions. Our responses to the Reviewers’ comments are detailed point-by-point below. 

We hope that all concerns have been adequately addressed and that the revised manuscript is now suitable for publication in PLOS ONE. Please do not hesitate to contact us if you require further information. 

Reviewer #1: Main summary

In the paper by Bumdelger et al, the authors evaluate the role of aneurysm/dissection in the Apoe-/- osteoprotegerin (Opg)-knockout mouse model. Interestingly, the authors introduce that using the calcium-chloride model, that deficiency of Opg had larger aneurysms and increased expression of matrix metalloproteinase-9 (MMP9) and TNF-related apoptosis-inducing ligand (TRAIL). Within this paper, using 7-month-old mice, AngII-infusion into Apoe-/- Opg-/- male mice resulted in lower external abdominal aortic diameter and lower incidence of aneurysms and dissections. The group goes on to do histological analysis and finds that collagen I expression is higher in the Apoe-/- Opg-/- mice under AngII-infusion, however this did not occur under vehicle conditions (water-infused, Supplemental Fig. 3). The group looks at alpha-smooth muscle actin, vimentin, TRAIL, F4/80, and collagen I and find that within mice that experience an aneurysm, Apoe-/- Opg-/- have increased expression of actin, vimentin, TRAIL, and collagen I that results in fibrotic remodeling that may help to lower the possibility of developing a dissection or aortic rupture. While the studies are interesting, there are some details that need to be addressed for these experiments.

Major concerns

1. If I understand correctly, both with calcium-chloride and the AngII-infusion model, TRAIL levels go up significantly. The authors state that TRAIL goes up, because the decoy receptor (Opg) is no longer expressed. If this is the case, then immunostaining for TRAIL should also be increased in the water-infused mice. If TRAIL expression is only tied to Opg, then the reader would benefit from knowing this in the water-infused mice. This could be placed in the supplement and help with the overall understanding of TRAIL within these models.

We thank the Reviewer for the suggestion. We did not detect significant differences in Trail expression between H2O-infused ApoE-/-Opg+/+ and ApoE-/-Opg-/- mice. However, AngII infusion significantly increased Trail expression only in ApoE-/-Opg-/- mice. These data were added to Supplemental Figures 5D and 5F. 

2. The authors do a good job of separating out non-aneurysm from aneurysm and dissection. In Figures 1-3, the authors examine all disease types, however in Figure 4, only aneurysm tissue is examined. Is it possible to look at non-aneurysm and dissection tissues for alpha-actin, vimentin, F4/80 and collagen I? It might help to keep it all consistent from figure to figure. There was also quantification for Figure 3, but none for Figure 4. Is it possible to do this for only 1 of the 4 proteins observed? Maybe vimentin or smooth muscle actin since quantification has already been done for collagen I.

We thank the Reviewer for the suggestion. As suggested, we performed IHC studies using antibodies against α-SMA, vimentin, and F4/80 for all disease types and also quantified the α-SMA expressing area in the adventitia. These data were added to Figure 4 (former Figure 4 has been renumbered to be Figure 5). Corresponding changes were also made to the manuscript (page 7, lines 18-19). 

3. Have the authors evaluated collagen IV levels in the aortic basal lamina? Is it increased in the Opg-knockout similar to collagen I?

Unfortunately, we did not perform that experiment. We did, however, perform IHC studies using an anti-collagen III antibody and found that collagen III was not expressed in the vascular wall of both ApoE-/-Opg+/+ and ApoE-/-Opg-/- mice. These data were added to Supplemental Figure 4B. 

4. It might be interesting to look at MMP2 and MMP9 levels in these Opg-knockout mice. My guess is that this might be lower in the Opg-knockout mouse, but if there is excessive remodeling off the tissue, then maybe not? This could be explored at the mRNA level if needed. Methods indicate that mRNA was collected, however I could not find any data concerning this.

We performed quantitative RT-PCR to measure Mmp2 and Mmp9 transcript levels. No significant differences were detected in Mmp9 levels between ApoE-/- Opg+/+ and ApoE-/- Opg-/- mice, although Mmp2 levels were elevated in ApoE-/- Opg-/- mice. Interestingly, AngII infusion did not influence Mmp2 levels, implying that either Mmp2 or Mmp9 is not involved in the histological changes induced by AngII infusion. These data were included in Supplemental Figures 5A-5C. 

Minor concerns

1. Was any ultrasound data collected? If it was collected, then placing this within the manuscript would benefit the reader.

We appreciate the Reviewer’s suggestion. However, we unfortunately did not collect ultrasound data. 

2. Methods state that the mice were aged out to 6 months, however abstract indicates 7 months. Which is correct?

We thank the Reviewer for pointing this out. The manuscript was revised to clarify that AngII infusions started when mice were 6 months of age. Mice were sacrificed at 7 months of age. 

3. Suprarenal aorta (SRA) appears in the Introduction first and should be defined there.

Suprarenal aorta (SRA) is defined in the manuscript on page 5 (line 5). We also added a representative image of the aorta to help explain the portion to which the SRA corresponds (Supplemental Figure 1). 

4. “Revealed” is misspelled in the Introduction.

This typographical error was corrected, accordingly (page 4, line 9).

5. Statistical analysis should be done on the incidence data in Figure 1.

Statistical analysis was performed for Figure 1C. We used ordinal logistic regression analysis. 

6. Did the authors measure any plasma/serum measures in the mice. Since these mice would be hypercholesterolemic, did the authors look at serum cholesterol levels?

Cholesterol concentrations were available and added to Supplemental Figure 2G. 

7. Did the authors look at TGF-beta or FGF in the aorta? Since the authors conclude that fibrosis remodeling has occurred in the Opg-knockout, then measurement of these factors could help to enhance the overall conclusion.

We thank the Reviewer for the helpful suggestion, and agree that Tgf-β may have an effect on fibrosis remodeling. We measured Tgf-β1 levels by RT-PCR and found that they increased after AngII infusion, consistent with a role for Tgf-β1 in enhancing fibrosis remodeling. We added an explanation of this to the manuscript (page 7, lines 10-13), and included the data in Figure 3C. 

8. There doesn’t seem to be any rupture information within these studies. Can the authors report on how many mice were lost due to aortic rupture in the wildtype and knockout mice?

Unfortunately, we only counted the number of dead mice and did not confirm whether their deaths were due to aortic rupture. Supplemental Figure 2F provides the survival data.

 

Reviewer #2: The paper entitled "Disruption of Osteoprotegerin has complex effects on medial destruction and adventitial fibrosis during mouse abdominal aortic aneurysm formation" by Bumdelger et al examined the effect of osteoprotegerin (Opg) in abdominal aneurysm. While there is already conflicting data published on the role of Opg on aneurysm: one by the same authors concluding preventive role of Opg in CaCl2-induced AAA and another by Moran et al 2014 concluding disease promoting role of Opg in AngII-induced AAA; this paper attempts to confirm the role of Opg in AngII-induced mouse model of AAA. However, at current state of the paper, many conclusions are not fully supported by the data provided. I have following concerns:

1) Figure 1: The title of Figure 1 and conclusion "Opg deficiency limits AngII-induced aortic dissection and dilatation" is overstated. External diameter is not significantly reduced with Opg deficiency. How do the authors define dissection? The authors need to classify aneurysm based on Daugherty's classification (PMID: 11606327). Were they abdominal or thoracic dissection? The data is not statistically analyzed to say preventive effect of Opg deficiency on aortic dissection. What about the mortality rates and mortality data in these experimental mice?

We thank the Reviewer for these insightful comments. As suggested, we revised the title and first paragraph of the Results section to be more commensurate in scope with the data. 

We did not initially follow Daugherty’s classification, since Moran’s group did not use that classification system. Nonetheless, we newly evaluated aneurysms based on the presence or absence of hematoma by morphological observation, similar to Daugherty’s classification. We cite Daugherty’s article and clarify that our classification method is a modified version of Daugherty’s method (page 5, lines 14-17). Our focus was mainly abdominal dissection at the SRA, although dissection was also observed in the thoracic region. 

As mentioned by Reviewer #1, we statistically analyzed the incidence rate of three disease phases using ordinal logistic regression. In result, there was a non significant tendency for aneurysm and aortic dissection in ApoE-/- Opg-/- mice. We included the mortality data in Supplemental Figure 2F.

2) Figure 2: I could not understand the mechanism behind adventitial thickening and aortic dissection. Do the authors have any speculation on how does the adventitial thickening leads to smaller aortic diameter and dissection in ApoE-/-Opg-/- mice?

To address this comment, we provide a schematic and description of how we believe adventitial thickening leads to smaller aortic diameter and dissection in ApoE-/-Opg-/- mice (Supplemental Figure 6). 

We speculate that administration of AngII in the ApoE-KO mouse model leads to inflammation, destruction of aortic tissue, and expression of inflammatory cytokines. In Opg deficient mice, AngII induces the expression of Trail, which may stimulate the appearance, proliferation, and migration of myofibroblasts, as indicated in previous reports. This accumulation of myofibroblasts may result in adventitial thickening with fibrosis, e.g., as a result of collagen I deposition. A substantial amount of collagen I in the adventitia could suppress dilatation of the inner diameter of the suprarenal aorta (SRA) by maintaining the stiffness and structural integrity of the lumen (Dobrin PB, et al. PMID: 7953454, Malfait F. PMID: 29709596). Thus, the characteristic changes in the adventitia of ApoE-/-Opg-/- mice, including the accumulation of myofibroblasts and collagen I, may underlie the suppression of aneurysm formation and obscure the influence of tissue destruction. Supporting this potential mechanism is a study by Dobrin et al., which found in ex vivo experiments that additional collagenase, rather than elastase, led to the enlargement of the lumen of blood vessels (Dobrin PB, et al. PMID: 7953454).

3) Figure 3: What about collagen III? Picrosirius red staining needs to be performed for differential staining of collagen I vs collagen III.

As suggested, we performed IHC experiments with antibodies against collagen type III. The expression of collagen III was much lower than that of collagen I and was not upregulated in response to AngII stimulation. We confirmed that the anti-collagen III antibody works by using a section of skin tissue as a positive control (Supplemental Figure 4B). 

4) Figure 4. The authors show that cells accumulated in adventitial region of Apoe-/-Opg-/- mice are likely myofibroblasts since they co-express SMA and vimentin. However, the Figure 4I shows more infiltration of F4/80+ macrophages in Apoe-/-Opg-/- compared to Apoe-/- alone in 4F. And it is surprising that there are fewer macrophages and less adventitial thickening in Apoe-/- mice in 4F and 4E. How does increased Trail expression, increased inflammation, myofibroblast accumulation and fibrotic remodeling can be protective to aortic dilation and dissection? The data presented and conclusion do not match. Instead, these data imply that there would be more disease in Apoe-/-Opg-/- mice because of more inflammation.

We describe the mechanism by which we speculate Trail expression is increased, inflammation is stimulated, myofibroblasts accumulate, and how fibrosis could suppress aortic dilatation and dissection in Supplemental Figure 6.

In further experiments, we detected an increase in macrophages, but no upregulation of Mmps, suggesting that inflammation does not play a major role in the observed histological changes. On the other hand, the thickened adventitia was filled with myofibroblasts and had substantial collagen I accumulation. As mentioned above in response to point #2, the presence of collagenase, rather than elastase, has been shown to cause enlargement of the lumen of blood vessels in a previous ex vivo study (Dobrin PB, et al. PMID: 7953454). These data indicate that fibrotic remodeling can be protective against aortic dilatation and dissection. 

5) Why 6-months (24 weeks) old mice are used for the aneurysm studies? Aneurysm studies are best performed in 8-12 weeks old mice.

Our initial goal was to reproduce the experiments performed by Moran’s group, and thus matched the ages of mice with their study. 

6) Poor quality of writing: There are several grammatical, typo and spelling errors in the sentences. The paper needs significant English editing.

We had the manuscript checked and edited by a native English speaker.

 

Reviewer #3: The paper entitled " Disruption of Osteoprotegerin has complex effects on medial destruction and adventitial fibrosis during mouse abdominal aortic aneurysm formation" by Kokubo et al concluded that fibrotic remodeling of the aorta induced by myofibroblast accumulation might be an important pathological event which limits AngII-induced aortic dilatation in ApoE -/- Opg -/- mice.

The conclusion of the manuscript is not substantially based upon the experimental data and is highly speculative.

Specifically:

1. The statement ‘Opg deficiency limits AngII-induced aortic dissection and dilatation’ is an overstatement of the data. The AAA definition is missing and also lacks statistical analysis.

We thank the Reviewer for raising this point. We agree and have revised the manuscript to avoid overstating the present findings (page 5, line 2, and title) and to include the definition of AAA (page 13, lines 17-19). 

2. No evidence to show the specificity of antibodies to detect vSMCs and fibroblasts. Single IHC may not be sufficient to draw the conclusions.

In the previous version of the manuscript, myofibroblasts were identified by IHC using anti-α-SMA or anti-vimentin antibodies. However, we agree that single IHC data cannot be used definitively to identify myofibroblasts. Accordingly, we performed immunofluorescence-based double staining with anti-α-SMA and anti-vimentin antibodies. Cells which were positive for both markers were considered to be myofibroblasts, and cells positive only for α-SMA were considered to be vSMCs (Skalli O, et al. PMID: 2918221).

3. The interpretation that ‘Opg deficiency was found to limit AngII-induced aortic dissection and dilatation’ is not convincing.

We added a discussion of how we arrived at that interpretation in Supplemental Figure 6 (and its accompanying legend). As mentioned in our reply to a similar comment from Reviewer #2, we believe that adventitial thickening accompanying collagen I accumulation has a preventive effect on enlargement of the aortic lumen and total outer diameter. According to a previous study, it was found in ex vivo experiments that additional collagenase, rather than elastase, led to the enlargement of the lumen of blood vessels (Dobrin PB, et al. PMID: 7953454). When this finding is applied to the present context, the accumulation of collagen I could be one of the causes underlying the preventive effect on enlargement of the aortic lumen.

4. There are no reports/data to show that fibrotic remodeling of the aorta might protects against AAA.

We agree with the Reviewer that there is no publication directly on point. However, as mentioned in the reply to the previous comment, Dobrin et al. found that a reduction in collagen I decreases the stiffness of the vascular wall (Dobrin PB, et al. PMID: 7953454). Moreover, according to a study by Malfait et al., corruption of collagen promoted the formation of aneurysms and dissection (Malfait F. et al. PMID: 29709596). Taken together, we believe our speculation that collagen I accumulation in the adventitia results in increased stiffness of the vascular wall to be reasonable. 

5. Why the authors used 6 month old mice?

Our initial goal was to reproduce the experiments conducted by Moran’s group. Accordingly, we used the experimental conditions described in their paper, including the age of mice.

6. Since aortic dissection in these mice occur at early stage of the disease (4-10 days), determination of visible blood at day 28 may not be a good indicator for aortic dissection.

As mentioned above, our initial goal was to reproduce the experiments conducted by Moran’s group. To this end, we mirrored the conditions they used, including those for observation period (i.e., 28 days after AngII infusion). Accordingly, we did not perform earlier stage observations. Classification of aneurysms by Daugherty’s group was also performed 28 days after AngII infusion, which is when the severity of aneurysms shows diversity in this AAA model (Daugherty A, et al. PMID: 11606327). Based on the above, we believe 28 days to be appropriate for observing the phenotype of mice. 

7. Please check for grammatical errors.

 We had the manuscript checked and edited by a native English speaker.

---

## [Decision Letter · Decision Letter 1]

8 Jun 2020

PONE-D-20-04332R1

Disruption of Osteoprotegerin has complex effects on medial destruction and adventitial fibrosis during mouse abdominal aortic aneurysm formation

PLOS ONE

Dear Dr. Kokubo,

Thank you for submitting your manuscript to PLOS ONE. After careful review, we consider it to have merit but it does not fully meet PLOS ONE’s publication criteria as it currently stands. Therefore, we invite you to submit a revised version of the manuscript that addresses the points raised during the review process.

Please respond to all the comments and provide additional information on antibodies and primers used in your study.

We look forward to receiving your revised manuscript.

Kind regards,

Helena Kuivaniemi, MD, PhD

Academic Editor

PLOS ONE

Additional Editor Comments (if provided):

Please provide tables in the supplementary material with details on the antibodies (vendor, cat#, lot#, specificity , literature citation using the same antibody) and primers (sequence, annealing temperature, PCR product size) used for the study

Reviewers' comments:

Reviewer's Responses to Questions

**Comments to the Author**

1. If the authors have adequately addressed your comments raised in a previous round of review and you feel that this manuscript is now acceptable for publication, you may indicate that here to bypass the “Comments to the Author” section, enter your conflict of interest statement in the “Confidential to Editor” section, and submit your "Accept" recommendation.

Reviewer #1: (No Response)

Reviewer #2: All comments have been addressed

Reviewer #3: All comments have been addressed

2. Is the manuscript technically sound, and do the data support the conclusions?

Reviewer #1: Yes

Reviewer #2: Yes

Reviewer #3: Yes

3. Has the statistical analysis been performed appropriately and rigorously? 

Reviewer #1: Yes

Reviewer #2: Yes

Reviewer #3: N/A

4. Have the authors made all data underlying the findings in their manuscript fully available?

Reviewer #1: Yes

Reviewer #2: Yes

Reviewer #3: Yes

5. Is the manuscript presented in an intelligible fashion and written in standard English?

Reviewer #1: Yes

Reviewer #2: Yes

Reviewer #3: Yes

6. Review Comments to the Author

Reviewer #1: Please make the following small editorial changes.

1. Hematoma is misspelled on page 13, second to last line.

2. Catalog numbers for all antibodies should be included to help with replication of studies.

3. Need a hyphen for alpha-SMA in Figure 4 legend

4. Need MMP2 primer sequence in the Methods section.

5. Should state both "MMP-2 and MMP-9" in Discussion section, page 10, second to last line.

Reviewer #2: All the comments have been nicely addressed by the authors with addition of data. I have no further questions.

Reviewer #3: All the comments have been addressed adequately by the authors. The reviewer has no more major concerns.

7. PLOS authors have the option to publish the peer review history of their article (what does this mean?). If published, this will include your full peer review and any attached files.

Reviewer #1: No

Reviewer #2: No

Reviewer #3: Yes: Chetan P Hans

---

## [Author Response · Author response to Decision Letter 1]

16 Jun 2020

Re: PONE-D-20-04332R1

Additional Editor Comments (if provided):

Please provide tables in the supplementary material with details on the antibodies (vendor, cat#, lot#, specificity , literature citation using the same antibody) and primers (sequence, annealing temperature, PCR product size) used for the study

Reviewers' comments:

Reviewer's Responses to Questions

Comments to the Author

1. If the authors have adequately addressed your comments raised in a previous round of review and you feel that this manuscript is now acceptable for publication, you may indicate that here to bypass the “Comments to the Author” section, enter your conflict of interest statement in the “Confidential to Editor” section, and submit your "Accept" recommendation.

Reviewer #1: (No Response)

Reviewer #2: All comments have been addressed

Reviewer #3: All comments have been addressed

2. Is the manuscript technically sound, and do the data support the conclusions?

Reviewer #1: Yes

Reviewer #2: Yes

Reviewer #3: Yes

3. Has the statistical analysis been performed appropriately and rigorously? 

Reviewer #1: Yes

Reviewer #2: Yes

Reviewer #3: N/A

4. Have the authors made all data underlying the findings in their manuscript fully available?

Reviewer #1: Yes

Reviewer #2: Yes

Reviewer #3: Yes

5. Is the manuscript presented in an intelligible fashion and written in standard English?

Reviewer #1: Yes

Reviewer #2: Yes

Reviewer #3: Yes

6. Review Comments to the Author

Reviewer #1: Please make the following small editorial changes.

1. Hematoma is misspelled on page 13, second to last line.

This typographical error was corrected accordingly.

2. Catalog numbers for all antibodies should be included to help with replication of studies.

We added a table in the manuscript to show catalog numbers for all antibodies followed by reviewer’s suggestion. We also changed text to cite this table in the section fo materials and methods, pp15 line 5-6. 

3. Need a hyphen for alpha-SMA in Figure 4 legend

We deleted α in the Figure 4 legend (page 23) because we have defined the α-smooth muscle actin as SMA in page 4, line 14. 

4. Need MMP2 primer sequence in the Methods section.

The MMP2 primer sequence has already been added in the method section (page 15, last line). To avoid confusion, we added an additional table in the manuscript to provide details on primers we used. We also changed text to cite this table in the section fo materials and methods, pp15 line 15.

5. Should state both "MMP-2 and MMP-9" in Discussion section, page 10, second to last line.

As followed reviewer’s suggestion, we stated both MMP-9 and MMP-2. 

Reviewer #2: All the comments have been nicely addressed by the authors with addition of data. I have no further questions.

Reviewer #3: All the comments have been addressed adequately by the authors. The reviewer has no more major concerns.

7. PLOS authors have the option to publish the peer review history of their article (what does this mean?). If published, this will include your full peer review and any attached files.

Do you want your identity to be public for this peer review? For information about this choice, including consent withdrawal, please see our Privacy Policy.

Reviewer #1: No

Reviewer #2: No

Reviewer #3: Yes: Chetan P Hans

---

## [Editor Report · Decision Letter 2]

18 Jun 2020

Disruption of Osteoprotegerin has complex effects on medial destruction and adventitial fibrosis during mouse abdominal aortic aneurysm formation

PONE-D-20-04332R2

Dear Dr. Kokubo,

We’re pleased to inform you that your manuscript has been judged scientifically suitable for publication and will be formally accepted for publication once it meets all outstanding technical requirements.

Congratulations!

kind regards,

Helena Kuivaniemi, MD, PhD

Academic Editor

PLOS ONE
---

## [Editor Report · Acceptance letter]

22 Jun 2020

PONE-D-20-04332R2 

Disruption of *Osteoprotegerin* has complex effects on medial destruction and adventitial fibrosis during mouse abdominal aortic aneurysm formation 

Dear Dr. Kokubo:

I'm pleased to inform you that your manuscript has been deemed suitable for publication in PLOS ONE. Congratulations! Your manuscript is now with our production department. 

Kind regards, 

on behalf of

Professor Helena Kuivaniemi 

Academic Editor

PLOS ONE